

# Effects of personal relevance and simulated darkness on the affective appraisal of a virtual environment

Alexander Toet[1], Joske M. Houtkamp[2,3] and Paul E. Vreugdenhil[2]

[1] TNO, Soesterberg, Netherlands
[2] Department of Information and Computing Sciences, Utrecht University, Utrecht, Netherlands
[3] Spatial Knowledge Systems, Alterra, Wageningen University and Research Center, Wageningen, Netherlands

## ABSTRACT

This study investigated whether personal relevance influences the affective appraisal of a desktop virtual environment (VE) in simulated darkness. In the real world, darkness often evokes thoughts of vulnerability, threat, and danger, and may automatically precipitate emotional responses consonant with those thoughts (fear of darkness). This influences the affective appraisal of a given environment after dark and the way humans behave in that environment in conditions of low lighting. Desktop VEs are increasingly deployed to study the effects of environmental qualities and (architectural or lighting) interventions on human behaviour and feelings of safety. Their (ecological) validity for these purposes depends critically on their ability to correctly address the user's cognitive and affective experience. Previous studies with desktop (i.e., non-immersive) VEs found that simulated darkness only slightly affects the user's behavioral and emotional responses to the represented environment, in contrast to the responses observed for immersive VEs. We hypothesize that the desktop VE scenarios used in previous studies less effectively induced emotional and behavioral responses because they lacked personal relevance. In addition, factors like signs of social presence and relatively high levels of ambient lighting may also have limited these responses. In this study, young female volunteers explored either a daytime or a night-time (low ambient light level) version of a desktop VE representing a deserted (no social presence) prototypical Dutch polder landscape. To enhance the personal relevance of the simulation, a fraction of the participants were led to believe that the virtual exploration tour would prepare them for a follow-up tour through the real world counterpart of the VE. The affective appraisal of the VE and the emotional response of the participants were measured through self-report. The results show that the VE was appraised as slightly less pleasant and more arousing in simulated darkness (compared to a daylight) condition, as expected. However, the fictitious follow-up assignment had no emotional effects and did not influence the affective appraisal of the VE. Further research is required to establish the qualities that may enhance the validity of desktop VEs for both etiological (e.g., the effects of signs of darkness on navigation behaviour and fear of crime) and intervention (e.g., effects of street lighting on feelings of safety) research.

Corresponding author
Alexander Toet, lextoet@gmail.com

## INTRODUCTION

This study investigated whether personal relevance influences the affective appraisal of a desktop virtual environment (VE) representing a prototypical Dutch polder landscape in different simulated (daytime and nighttime) lighting conditions. We use the term affective appraisal (*Russell & Lanius, 1984*; *Russell & Snodgrass, 1987*) to refer to emotional appraisals of an environment, to make a clear distinction with emotional response to that environment. Affective appraisals are the attributed emotional or affective qualities, or cognitions about possible object- or place- elicited holistic percepts (*Russell & Snodgrass, 1987*). Affective appraisal measures the evaluation of the significance of the environment for our personal wellbeing. Affective responses (emotion and mood) are the result of appraisal and occur if the environment is judged relevant for our goals and wellbeing (*Lazarus, 1991*).

Nighttime outdoor environments are typically appraised as less pleasant and more frightening than their daytime equivalents (*Bishop & Rohrmann, 2003*; *Loewen, Steel & Suedfeld, 1993*). In the real world, ambient darkness evokes feelings of fear for personal safety (*Box, Hale & Andrews, 1988*; *Cozens, Neale & Hillier, 2003*; *Nasar & Jones, 1997*) and determines human (navigation) behavior (*Warr, 1990*), particularly in the absence of social presence (*Painter, 1996*). Ambient darkness elicits fear by concealing potential or imagined dangers (*Blöbaum & Hunecke, 2005*; *Gray, 1987*; *Lyons, 1980*; *Nasar & Jones, 1997*; *Warr, 1990*) and can turn places that are pleasant during daylight into frightening places after dark (*Hanyu, 1997*; *Nasar & Jones, 1997*). As a result, many people (especially women) avoid leaving home or visiting certain places after dark (e.g., *Fisher & Nasar, 1992*; *Keane, 1998*; *Warr, 1985*). Interventions like environmental design (*Cozens & Love, 2015*), lighting improvements (*Fotios, Unwin & Farrall, 2015*; *Painter, 1996*) and intelligent street lighting (*Haans & de Kort, 2012*; *Van Rijswijk, Haans & De Kort, 2012*) may help to reduce fear and improve street use at night. VEs may be cost effective tools to design, evaluate and optimize such interventions (*Boomsma & Steg, 2014a*; *Boomsma & Steg, 2014b*; *Cozens, Neale & Hillier, 2003*; *Nikunen & Korpela, 2012*). However, their suitability for this purpose depends critically on their ability to correctly address the user's affective, cognitive and perceptual experience (*Lewis, Casello & Groulx, 2012*; *Wergles & Muhar, 2009*). This means that the affective appraisal of a VE should vary with ambient lighting in the same way as those of a similar real counterpart. In other words, a nighttime VE should evoke the same (affective and behavioral) responses as a similar nighttime real environment (i.e., the VE should be ecologically valid).

The ecological validity of *immersive* daytime VEs for the study of feelings of fear and their impact on human navigation behavior in built environments has convincingly been demonstrated (e.g., *Park et al., 2008*; *Park et al., 2010*; *Park et al., 2012*; *Park et al., 2011*). Also for an immersive system, it has been shown that simulated driving through dark virtual tunnels induces ecologically valid negative affect and corresponding startle responses (*Mühlberger, Wieser & Pauli, 2007*). In contrast to immersive systems, the ecological validity of *desktop* (i.e., non-immersive) VEs for evoking darkness related emotional and behavioral responses is still unresolved. Commercial desktop video games often use darkness in an

attempt to evoke suspense and dread (e.g., Slender: www.slendergame.com, The Suffering: Midway Games, Silent Hill 2: Konami; see also *El-Nasr, 2006*; *Niedenthal, 2005*). Darkness is indeed one of the most often reported causes of fear by video game players (*Lynch & Martins, 2015*). Although perceived safety of VEs decreases with reduced lighting levels in a similar manner as in real environments (*Boomsma & Steg, 2014a*; *Boomsma & Steg, 2014b*), previous studies observed only a small effect of simulated darkness in desktop VEs on the user's behavioral and emotional responses (e.g., *Rohrmann & Bishop, 2002*; *Toet, Van Welie & Houtkamp, 2009*).

There may be several reasons why previous studies failed to find larger effects of simulated darkness on the effective appraisal of desktop VEs. *Rohrmann & Bishop (2002)* compared the affective appraisal of the daytime and nighttime versions of a simulated suburban environment. Their participants watched video clips showing walkthroughs of the VE and judged their liking and appreciation of the area and their personal safety related associations. They rated the nighttime VE as more threatening and arousing than its daytime equivalent. However, the overall threat ratings were below neutral (i.e., the environment was simply not perceived as very threatening or arousing in any of the tested lighting conditions). The fact that the nighttime VE was not considered very threatening may be a result of the fact that the overall light level in the nighttime VE was still sufficient to get a good overview of the environment and the fact that the soundtrack (sounds of passing traffic and footsteps) suggested social presence. Both factors probably had a reassuring influence on the participants. *Bishop & Rohrmann (2003)* compared the affective appraisal of a real urban park area with that of its simulated counterpart, both for daylight and nighttime conditions. Their participants either performed a walkthrough of the real environment (either in daytime or at night) or watched a video clip of a walkthrough of the simulated environment (shown either in simulated daylight or darkness). Although the real and virtual environments were both perceived as less pleasant and more threatening at night, the differences in affective appraisal (i.e., differences in liking and appreciation of the area and personal safety related associations) were small. Again, this is probably due to a combination of a relatively high level of ambient lighting and social presence: the participants were walked through the real environment at night by the experimenter in groups of 10 (most likely resulting in a strong sense of social presence), while the relatively well-lit VE also contained signs of social presence (trams, cars, sounds). In a previous study (*Toet, Van Welie & Houtkamp, 2009*) we compared the affective appraisal of a desktop VE representing an old Italian village both for simulated day- and nighttime conditions. We found only a small effect of simulated darkness on the affective appraisal of the VE: observers appraised the nighttime version of the VE only slightly less pleasant and more arousing than its daytime equivalent. We attributed this weak effect to the fact that the VE had a cosy atmosphere, sufficient lighting to distinguish most details of the environment, and a soundtrack that suggested social presence (music, people singing, murmuring voices, etc.). *Kim et al. (2014)* compared people's fear rating and eye movements in response to both actual nighttime outdoor environments and to images of the same scenes. While their participants' eye fixation behavior was similar in both conditions, the image-based environments were rated overall as less frightening than the

actual environments. Summarizing, although previous studies showed that images and desktop simulation of outdoor nightscapes are appraised as somewhat less pleasant and more frightening than their daytime counterparts, the overall effects were small, probably due to ameliorating factors like social presence and relatively high levels of ambient lighting.

An important factor that has not been investigated in previous studies is the *personal relevance* of a simulation. Personal relevance is the extent to which a VE itself or actions therein have real personal consequences and/or intrinsic importance for the user. It is known that events or situations that are appraised as relevant and significant to one's goals and wellbeing induce emotions more effectively than irrelevant ones (*Freeman et al., 2005*; *Lazarus, 1991*). For instance, the emotional valence of visual scenes is significantly enhanced when they are paired with short sentences inducing self-reference (e.g., "*this dog will attack you*" written underneath the image of an aggressive dog: *Walla et al., 2013*). Simulations are also more likely to affect the user's emotional state when they have a higher degree of personal relevance (*Freeman et al., 2005*; *Hoorn, Konijn & Van der Veer, 2003*). It has even been argued that presence and emotions may be induced more effectively by enhancing the personal relevance of a VE rather than by increasing its perceptual realism (*Hoorn, Konijn & Van der Veer, 2003*). A result that appears to confirm this hypothesis is the finding that the perceived risk of a health message presented in a virtual environment is effectively increased when it is delivered by the user's virtual doppelgänger (suggesting a direct link with one's own personal health: *Ahn, Fox & Hahm, 2014*). A lack of personal relevance may also explain why people experienced less fear in a virtual nighttime environment (no relevance for one's wellbeing) than in its real-world counterpart (direct relevance for one's wellbeing: *Kim et al., 2014*). Summarizing, it appears that—next to social presence and relatively high levels of ambient lighting—a lack of personal relevance may have been a fear-reducing factor in most previous studies investigating the effects of simulated darkness in desktop VEs on human emotion and behavior. Hence, a lack of personal relevance in these studies may have caused their finding that darkness related feelings of fear induced by desktop VEs were much weaker than the feelings of fear that people experience in similar real world conditions.

This study investigates if personal relevance can intensify the affective appraisal of a desktop VE in simulated darkness. The VE represents a prototypical deserted Dutch rural area. Participants were requested to explore either a daytime or a nighttime version of this VE. We selected an environment with sufficient prospect (open spaces; low entrapment) since lighting is known to affect feelings of safety most strongly in this type of environments (*Blöbaum & Hunecke, 2005*). The only illumination provided in the nighttime version of this VE originated from some scattered streetlights along the roads and stars in the partly clouded sky, resulting in a very dark environment. In addition, there were no signs of social presence. In some conditions the participants were led to believe that the virtual walking tour through the VE would prepare them for a tour through a similar real environment, either in the same or in the opposite lighting condition as presented the experiment. This fictional assignment served to enhance the personal relevance of the simulation. A combination of intense darkness, lack of social presence and enhanced personal relevance was used in an attempt to more effectively evoke darkness related feelings of fear. The

affective appraisal of the VE (in terms of atmospheric parameters, as detailed in the Measures section, and adapted from *Vogels, 2008a*) and the emotional state of the participants were measured through self-report. Based on the results of previous studies we expect that our desktop VE is appraised as less pleasant and more arousing in simulated darkness than in simulated daylight. Our main hypothesis is that increased personal relevance of the VE enhances this effect. More specifically, we expect (H1) that participants who explore the nighttime VE with the assignment to visit to the corresponding real world environment at night appraise the VE more negatively than participants without this assignment. In addition, we expect that personal relevance also affects both the emotional response to the VE and the participants' mood. That is, we expect that participants with the additional assignment experience both (H2) more intense short term (emotions) and (H3) longer lasting (mood) negative affective feelings than participants without this assignment.

Summarizing, our main hypotheses are that participants who explored the nighttime desktop VE with the information that this experience serves to prepare them for a walkthrough of the corresponding real environment by night (increased and negative personal relevance)

(H1)  rate the VE as (H1a) less *Cosy* and (H1b) more *Tense*,

(H2)  experience (H2a) less *Pleasure* and (H2b) more *Arousal*, and

(H3)  experience (H3a) a larger decrease in *Positive Affect* and (H3b) a larger increase in *Negative Affect* than participants without this information.

Finally, we hypothesized that (H4) participants with a real world follow-up assignment (increased personal relevance) experience a higher degree of presence in the VE than participants without this information.

## METHODS

### Participants

A total of 72 female volunteers, aged between 17 and 32 years ($M = 22.2$ years, SD $= 2.9$ years) participated in this experiment. A sample of young females was chosen because it is known that this group is particularly susceptible to fear of darkness (*Blöbaum & Hunecke, 2005*; *Loewen, Steel & Suedfeld, 1993*; *Warr, 1984*; *Warr, 1990*), and shows a greater risk awareness which also extrapolates to virtual environments (*Boomsma & Steg, 2014a*; *Park et al., 2012*). Participants were randomly allocated to one of the 6 experimental conditions, such that each condition was performed by 12 participants. The participants were students of the Utrecht University (Utrecht, The Netherlands) and were recruited by public announcements. The experiment was performed in accordance with the Helsinki Declaration of 1975, as revised in 2013 (*World Medical Association, 2013*), and ethical guidelines of the American Psychological Association. All participants gave their written consent. Each participant received an incentive of 10 Euros for taking part in the study.

### Experimental design

Participants explored either a daytime or a nighttime version of a desktop VE, and gave their affective appraisal and emotional response. In four conditions the participants were led to believe that the tour they were about to make through the VE actually would prepare

them for a follow-up tour through a similar real-world area, either in the same (daylight VE with daylight follow-up tour or nighttime VE with nighttime follow-up tour) or in opposite (daylight VE with nighttime follow-up tour or nighttime VE with daylight follow-up tour) lighting conditions as used in the simulation. This fictitious assignment served to increase the personal relevance of the simulation. As a result, the experiment had a $2 \times 3$ design: two simulated lighting conditions (daylight/darkness) and three fictitious follow-up assignment conditions (no assignment, or assignment related to either the same or opposite lighting conditions).

## Procedure

The timeline of the experimental procedure is shown in Fig. 1. After being welcomed to the lab, the participants first answered some demographic questions, and some questions to assess their propensity for fear of darkness in real-life and their gaming experience. Then, their emotional state was assessed for the first time through their responses to the PANAS questionnaire. Next, they read their instructions, which informed them that they were about to explore a virtual polder landscape for about 10 min, after which they would be asked to draw a map of the entire area, including the off-the-road parts. Participants in the fictitious assignment conditions were also asked to take part in a follow-up task, which involved a visit to the hypothetical real area corresponding to the simulation, either in daytime or at night. They were told that they would not receive any assistance during that visit, and that they would have to rely on their previous experience in the VE to perform the real world exploration task. Directly after reading their instructions the participants self-reported their current emotional state for the first time using the SAM. Then, the participants explored the VE for 10 min. Afterwards, they filled out the affective appraisal questionnaire, followed by the SAM and the PANAS (both for the second time), and the IPQ presence questionnaire. Next, all participants drew a map of the virtual environment. After drawing the map, they could give their comments about the experiment in response to an open question. During a debriefing at the end of the experiment the experimenter informed the participants about the real purpose of the experiment and asked them not to communicate this to future participants. The total duration of the experiment was about 35 min for each participant.

## Materials
### The virtual environment

The VE used in this study represents a prototypical Dutch polder landscape with some scattered houses, low-lying tracts of grasslands enclosed by dikes, roads, railway tracks, canals, and levees (see Fig. 2). It was originally developed as a training tool for levee patrollers by GeoDelft (now Deltares: www.deltares.nl) and Delft University of Technology, using the Unreal Engine 2 Runtime game engine (for full rendering details see: *Harteveld et al., 2007*). The simulation contains no people; only some birds flying around and several sheep in one of the grasslands. A soundtrack (representing wind and breaking waves) and visual dynamics (e.g., waving trees, water waves etc.) serves to enhance the realism and immersiveness of the simulation (*Houtkamp, Schuurink & Toet, 2008*). In the daytime condition the environment is lit by the sun. In the nighttime condition streetlights along

the roads and stars in the partly clouded sky provide the only illumination. We selected this environment since it is known that feelings of safety and human behavior vary most strongly with lighting levels in settings with low entrapment (access to refuge) and low concealment (open space; *Blöbaum & Hunecke, 2005*).

### Set-up

The simulation was performed on a Dell OptiPlex 755 desktop computer (www.dell.com) equipped with an Intel Core 2 Duo CPU, running at 2.99 Ghz, 1.96 GB RAM, a NVIDIA GeForce 8800GT graphics card (www.nvidia.com), and a standard mouse and keyboard. The simulated environment was displayed on a 22″ Dell E228WFP Flat Panel Color monitor. Sound was provided through an Altec Lansing ADA215 speaker set (www.alteclansing.com). The sound level was such that the sounds of the simulation were clearly audible and at a realistic level.

The entire set-up was placed in an artificially illuminated room. The windows were covered to block the sunlight. The lights were on (about 400 lux horizontal illumination) when the participants answered questionnaires or navigated through the daytime virtual environment. The lights were turned off (resulting in a dimly lit room with less than 100 lux horizontal illumination) when the participants navigated through the nighttime virtual environment. Since the lower light level was within the mesopic range—similar to most real world night-time outdoor scenes—the adaptation periods between both light levels was in the order of seconds (about 10 s: see *Adrian & Flemming, 1991*). Monitor settings were kept constant throughout the experiment.

Participants were comfortably seated at a distance of about 60 cm in front of the monitor. They used the mouse and keyboard to navigate through the VE. The experimenter was seated behind a second display placed on a desk at the other side of the room to the left side of the participant, where he could unobtrusively monitor the participant's actions.

### Map drawing

At the start of the experiment the participants were informed that they were required to draw a map of the simulated area after completing their virtual walking tour. This instruction served to stimulate the participants to actively explore most of the simulated area, so that they would not linger in one part. In addition, it served to confirm the fictitious follow-up assignment: the participants in that group were led to believe that they would be allowed to use the map they had drawn based on their exploration of the VE to find their way in the corresponding real environment at a later stage. The maps which the participants produced were not analyzed further in this study.

## Measures

This section briefly presents the questionnaires that were used in this study.

Experimental measures were questionnaires that measured respectively the participants' affective appraisal of the VE and their affective responses (emotions and mood). Questionnaires that measured respectively the participants' fear of darkness in real life,

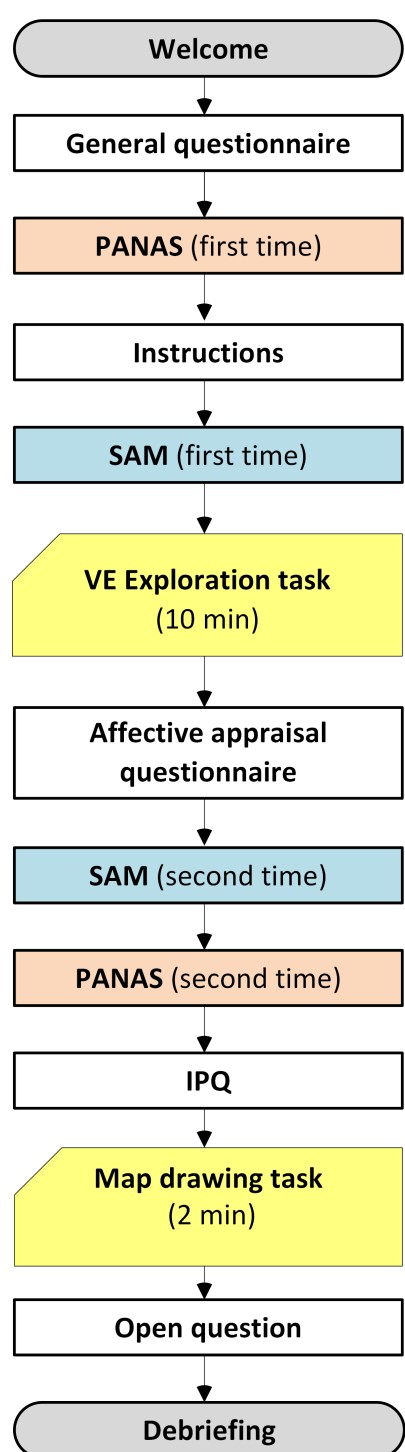

**Figure 1** Timeline of the experimental procedure.

their sense of presence in the VE, and their game and navigation experience, served as control measures.

For full details of these questionnaires see the Supplemental Information accompanying this paper.

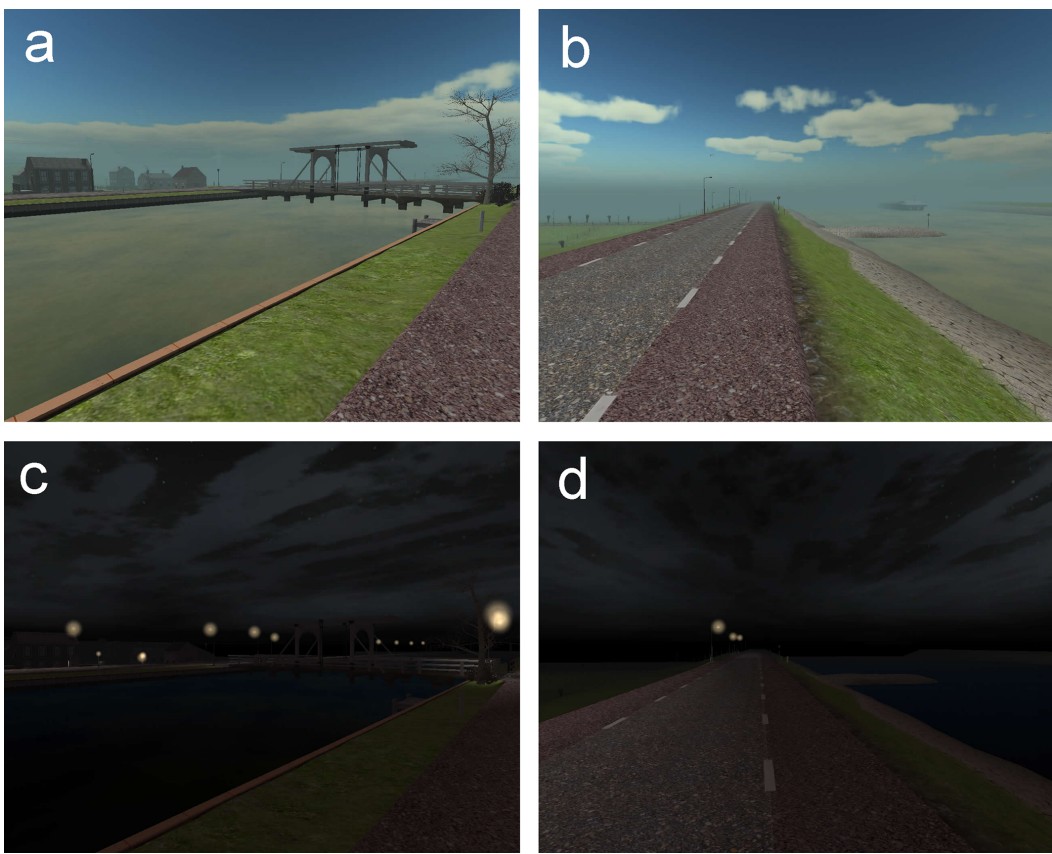

**Figure 2 Screenshots of the VE in daytime (A, B) and at night (C, D).**

### Experimental measures

*Affective appraisal.* To assess whether different degrees of lighting and personal relevance influenced the affective appraisal of the VE, we applied a subset of the 38 adjectives from a differential rating scale that was designed to assess the perceived atmosphere of built environments (*Vogels, 2008a*). In this context, atmosphere is defined as the affective evaluation of the environment. Atmosphere gives information about the expected effect of the environment on people's affective state (*Vogels, 2008b*). The 11 selected terms represent each of its four principal affective dimensions (*Vogels, 2008a*): Cosyness (*Cosy, intimate, safe*; in Dutch: *behaaglijk, intiem, veilig*), Liveliness (*lively, inspiring, stimulating*; in Dutch: *levendig, inspirerend, stimulerend*), Tenseness (*tense, terrifying, threatening*; in Dutch: *gespannen, beangstigend, bedreigend*), and Detachment (*business, formal*; in Dutch: *zakelijk, formeel*). Each term was scored on a 7-point rating scale ($-3 = $ *not at all*, $3 = $ *very much*).

*Emotional response.* We measured the participants' emotional response to the announcement of the fictitious follow-up task and to the experience of the VE, to assess whether these events influenced their affective state. Emotions are transient, relatively short lasting states of feeling, that are usually caused by the appraisal of the relevance of

specific event or environment for personal wellbeing: the more relevant an environment or event, the more emotive it can be. The participants self-reported their momentary feelings of pleasure, arousal and dominance using a validated 9-point pictorial rating scale (the Self-Assessment Manikin or SAM: *Bradley & Lang, 1994*). The SAM provides a simple, fast, and non-linguistic way of assessing emotional state along the principal emotional dimensions of Valence, Arousal and Dominance (*Mehrabian & Russell, 1974*), and is therefore highly suitable to measure transient (short term) emotional states. The SAM was developed by *Lang (1980)* as an efficient alternative to the Semantic Differential Scale created by *Mehrabian & Russell (1974)*. In this study, the SAM was applied twice: once just after the participants had read their assignment and before they started their tour through the virtual environment (to measure their emotional state directly after reading the task assignment), and once after they completed their virtual tour. This test served to check whether participants with a fictitious follow-up assignment (i.e., participants who believed they had to explore a similar real environment at a later stage) experienced emotions that were different from those experienced by participants who performed the experiment without this assignment.

*Mood.* Light and dark environments may induce different moods (a mood is a longer lasting affective state that can not necessarily directly be linked to a discrete stimulus or event). Mood was measured through self-assessment using a validated Dutch translation of the Positive and Negative Affect Scale (PANAS: *Watson, Clark & Tellegen, 1988*; for the translation see: *Engelen et al., 2006*; *Peeters, Ponds & Vermeeren, 1996*). This is a list of 20 adjectives used to describe different emotional states: 10 states of Positive Affect (PA) and 10 states of Negative Affect (NA). The PA scale measures activity and pleasure, while the NA scale relates to fear and stress. A high PA is a state of high energy, full concentration and pleasurable engagement whereas low PA is characterized by sadness and lethargy. The NA dimension is the degree of distress and unpleasant engagement. High NA implies anger, disgust, guilt, fear and nervousness while low NA implies calmness and serenity (*Watson, Clark & Tellegen, 1988*). Because of its length (and in contrast to the SAM) the PANAS is more suitable to measure longer lasting emotional states (moods). Participants scored the extent to which they experienced each emotional state on a 5-point unipolar rating scale (1 = *not at all or very slightly*, 5 = *extremely*).

*Presence.* In the context of simulation and gaming the term 'presence' usually refers to the subjective experience of 'being there' in the mediated environment (*Schuemie et al., 2001*; *Slater & Wilbur, 1997*). It appears that participants experience a higher sense of presence in a VE when they are emotionally affected by it (*Baños et al., 2004a*; *Baños et al., 2004b*; *Baños et al., 2006*; *Baños et al., 2008*; *Gorini et al., 2011*; *Riva et al., 2007*; for a recent review see: *Diemer et al., 2015*). It has also been observed that a VE scenario more effectively induces emotions when it has personal relevance (*Baños et al., 2004a*; *Freeman et al., 2005*). To assess whether the fictitious follow-up assignment (personal relevance) enhances feelings of presence, we used the Dutch translation of the Igroup Presence Questionnaire (IPQ, downloaded from http://www.igroup.org/pq/ipq; see *Schubert, Friedmann & Regenbrecht,*

*2001*). The IPQ contains 14 questions that are scored on a bipolar 7-point rating scale. These items load respectively on the factors Spatial Presence (SPR), Involvement (INV) and Realness (REA).

### Control measures

*Fear of darkness in the real world.* Simulated darkness in a VE is probably more likely to affect someone who is also affected by darkness in the real world. In the real world cues like darkness (day/night), novelty (familiar/unfamiliar) and lack of social presence are known to evoke fear of victimization and determine navigation behavior, especially in women (*Fisher & Nasar, 1992*; *Warr, 1984*; *Warr, 1990*). To check if our female volunteers also felt less comfortable in darkness in real life (and are therefore comparable on this aspect to populations observed in previous studies in the literature), we tested their susceptibility to each of these cues by scoring eight statements (*I'm very well able to find my way/in an unfamiliar environment/in a familiar environment at night/in an unfamiliar environment at night; I can orientate very well/in the dark/in daytime; I dare to walk by myself in an unfamiliar environment/at night/in daytime; I feel uncomfortable in the dark*) on a 7-point bipolar rating scale ($-3 =$ *strongly disagree*, $3 =$ *strongly agree*), prior to the main experiment.

*Game and navigation experience.* Problems with navigation can degrade the perceived realism of a simulation (*IJsselsteijn et al., 2000*). Since frequent game players probably have acquired higher levels of navigation proficiency, the navigation through the VE may require less of their attention so that they may achieve higher levels of presence. To control for this effect we measured game experience by two questions ("*How frequently do you play 3D computer games?*" and "*How frequently do you use other virtual environments (e.g., Second Life)?*"), using a 5-point unipolar rating scale ($1 =$ *never*, $5 =$ *very often*). In addition, the extent to which navigation in the present simulation required attention and interfered with task performance was measured after the exploration of the VE by two questions ("*Did you need your attention to navigate?*" and "*Did the navigation control hinder your task performance in the virtual environment?*") using a 5-point unipolar rating scale ($1 =$ *not at all*, $5 =$ *very much*).

*Open question.* The experiment ended with the question whether the participant had any comments about the experiment.

## Data collection and analysis

A web-based survey tool (http://www.surveymonkey.com) was used to apply all measures used in this study. The answers were stored online and were later uploaded for further analysis. All statistical analyses were performed with IBM SPSS 22.0 for Windows (www.ibm.com). For all analyses, a probability level of $p < .05$ was considered to be statistically significant.

**Table 1** **Affective appraisal of the VE in terms of *Cozyness, Liveliness, Tenseness* and *Detachment*.** Appraisals given by participants who explored either a daytime or nighttime VE with respectively no additional assignment, or with the suggestion that they would be asked to traverse a corresponding real environment during either daylight or darkness (fictitious follow-up assignment). $N = 12$ for each condition.

| Simulated lighting | Fictitious task | Cosiness | | Liveliness | | Tenseness | | Detachment | |
|---|---|---|---|---|---|---|---|---|---|
| | | M | SD | M | SD | M | SD | M | SD |
| | No task | 0.25 | 0.88 | −1.00 | 1.37 | −2.56 | 0.67 | −1.21 | 1.70 |
| Daylight | Daylight | 0.28 | 1.30 | −0.56 | 1.15 | −2.25 | 0.89 | −1.17 | 1.67 |
| | Darkness | 0.50 | 1.12 | −0.16 | 1.34 | −1.94 | 0.87 | −0.67 | 1.44 |
| | None | −0.78 | 1.04 | −0.53 | 1.41 | −0.42 | 1.31 | −1.29 | 1.05 |
| Darkness | Darkness | 0.06 | 0.91 | −0.50 | 0.83 | −0.61 | 1.29 | −0.83 | 1.23 |
| | Daylight | −0.75 | 1.02 | −0.42 | 0.91 | 0.06 | 1.32 | −0.92 | 1.40 |

## RESULTS

### Effects of personal relevance on environmental appraisal

The results of the affective appraisal questionnaire are listed in Table 1. The *Cosiness* of the daylight representation of the VE was rated above neutral for all conditions. In contrast, the nighttime VE was rated mostly negative or near neutral on *Cosiness*.

The data contained no outliers (from visual inspection of boxplots of the data) and the assumptions of normality (verified by a Shapiro–Wilk test) and sphericity (verified by Levene's test) were satisfied. We therefore performed a two-way independent ANOVA to assess the effects of lighting and follow-up assignment (personal relevance) on the appraisal of the environment.

The two-way independent ANOVA showed a main effect of ambient lighting on both *Cosiness* and *Tenseness*. *Cosiness* was rated significantly lower for the nighttime environment than for its daytime equivalent ($F(1,66) = 10.90$, $p = .002$, partial $\eta^2 = 0.142$), while the factor *Tenseness* was rated significantly more applicable to the nighttime VE than to its daytime counterpart ($F(1,66) = 56.16$, $p < .001$, partial $\eta^2 = 0.460$). This result confirms our assumption that the nighttime VE is indeed appraised more negatively in simulated darkness than in simulated daylight, and agrees with previous findings in the literature that desktop simulations of outdoor nightscapes are typically appraised as less pleasant and more frightening than their daytime counterparts.

The two-way independent ANOVA showed that the fictitious nighttime follow-up task in the real world had a small but significant effect on *Cosiness* for the nighttime VE ($F(1,22) = 4.381$, $p = .048$, partial $\eta^2 = 0.166$). Contrary to our hypothesis (H1a), the nighttime VE was appraised as slightly more *Cosy* with a real-world nighttime follow-up assignment compared to no assignment. The fictitious nighttime follow-up task had no effect on the factor *Tenseness* ($F(1,22) < 1$, n.s.). Thus, our main hypothesis (H1) that participants who explored the nighttime VE with the assignment to explore the corresponding real environment by night would rate the VE as (H1a) less *Cosy* and (H1b) more *Tense* than participants without this assignment, is not confirmed.
**Table 2  SAM scores (rated on a 9-point scale).** Pleasure, arousal and dominance were rated before (T1) and after (T2) the exploration of the VE.

| Simulated lighting conditions | Fictitious task | Pleasure T1 | | Pleasure T2 | | Arousal T1 | | Arousal T2 | | Dominance T1 | | Dominance T2 | |
|---|---|---|---|---|---|---|---|---|---|---|---|---|---|
| | | M | SD | M | SD | M | SD | M | SD | M | SD | M | SD |
| Daylight | No task | 6.50 | 1.24 | 5.42 | 1.93 | 3.17 | 1.12 | 2.58 | 1.51 | 6.00 | 1.95 | 6.17 | 2.04 |
| | Daylight | 6.67 | 1.16 | 6.17 | 1.70 | 3.17 | 1.59 | 2.75 | 1.60 | 5.25 | 1.55 | 5.00 | 1.28 |
| | Darkness | 6.83 | 0.94 | 6.25 | 1.49 | 2.83 | 1.03 | 2.92 | 1.73 | 5.42 | 1.56 | 5.67 | 1.61 |
| Darkness | No task | 6.92 | 1.38 | 6.25 | 1.49 | 3.00 | 1.54 | 3.50 | 1.31 | 5.58 | 1.88 | 5.50 | 2.28 |
| | Darkness | 5.42 | 1.68 | 5.25 | 1.66 | 3.25 | 1.55 | 3.58 | 1.51 | 4.73 | 2.15 | 5.27 | 1.45 |
| | Daylight | 6.75 | 0.62 | 5.17 | 1.27 | 3.58 | 1.56 | 3.83 | 1.34 | 5.58 | 1.31 | 5.17 | 1.47 |

The two-way independent ANOVA also showed that the fictitious (either nighttime or daytime) follow-up task in the real world also had no effect on both the *Cosiness* ratings and the *Tenseness* ratings ($F(1, 22) = 3.687$, $p = .068$) for the daytime VE (in all cases: $F(1, 22) < 1$, n.s.).

The factor *Liveliness* was rated negatively in all conditions, while the factor *Detachment* was rated consistently less than applicable to the VE in all conditions. A two-way independent ANOVA revealed no significant effects of ambient lighting and the fictitious follow-up task on these factors (in all cases: $F(1, 22) < 1$, n.s.).

Summarizing, the nighttime version of the VE was experienced as significantly less cosy and more tense than its daytime equivalent. Apart from the (somewhat surprising) effect that the nighttime VE was appraised as slightly more Cosy with a real-world nighttime follow-up assignment (compared to no assignment), we found no significant effects of a fictitious real-world follow-up task on the affective appraisal of the VE.

### Effects of personal relevance on emotional response

The factors *Pleasure*, *Arousal* and *Dominance* were rated using the SAM, just before the participants started their exploration of the VE (T1) and afterwards (T2). The results are shown in Table 2.

A Shapiro–Wilk test showed that the independent parameters (lighting condition and task) were not normally distributed. We therefore used a non-parametric Kruskal-Wallis test to analyze the results.

First we compared the *Pleasure* and *Arousal* ratings given by participants who explored the nighttime VE with the fictional follow-up real world nighttime assignment to the ratings given by participants without this follow-up task at T2 (just after exploring the VE). The results indicate that neither the *Pleasure* ($\chi^2(1, 22) = .986$, n.s.) nor the *Arousal* ($\chi^2(1, 22) = .127$, n.s.) ratings differ significantly between both groups. Thus, our hypothesis (H2) that participants who explored the nighttime VE with the fictional follow-up real world nighttime assignment experience (H2a) less *Pleasure* and (H2b) more *Arousal* than participants without this assignment, is not confirmed.

Next, we compared the *Pleasure* and *Arousal* ratings between the groups who respectively explored the nighttime and the daytime VE. Participants who explored the nighttime

**Table 3** **The mean and standard deviation of the ratings on the PANAS positive and negative affect scales.** Ratings were given before reading the instructions (T1) and after finishing the VE exploration task (T2).

| Simulated lighting | Fictitious task | PA (T1) | | PA (T2) | | NA (T1) | | NA (T2) | |
|---|---|---|---|---|---|---|---|---|---|
| | | M | SD | M | SD | M | SD | M | SD |
| | No task | 32.08 | 4.46 | 26.58 | 7.99 | 12.27 | 1.68 | 11.64 | 2.11 |
| Daylight | Daylight | 37.00 | 4.95 | 31.67 | 5.71 | 12.25 | 1.77 | 12.50 | 2.78 |
| | Darkness | 36.42 | 5.45 | 33.50 | 6.19 | 12.83 | 3.22 | 13.75 | 3.72 |
| | No task | 35.75 | 6.45 | 35.00 | 5.77 | 12.08 | 2.31 | 12.58 | 2.19 |
| Darkness | Darkness | 31.42 | 5.73 | 28.25 | 6.40 | 13.50 | 3.78 | 14.50 | 3.40 |
| | Daylight | 36.08 | 3.73 | 31.00 | 4.35 | 15.08 | 3.53 | 15.75 | 3.11 |

VE showed both a significantly larger increase in *Arousal* (the difference between the measurements at T1 and T2: $\chi^2(1, 44) = 4.989$, $p = .027$, $\eta^2 = 0.07$) and a higher level of *Arousal* just after experiencing the VE (at T2: $\chi^2(1, 44) = 7.457$, $p = .006$, $\eta^2 = 0.11$) compared to participants who explored the daytime VE. No significant effect was observed for *Pleasure*. Hence, this result only partly agrees with previous findings in the literature that desktop simulations of outdoor nightscapes are appraised as less pleasant and more frightening than their daytime counterparts.

There were no significant differences between the SAM parameters in any of the other experimental conditions.

Summarizing, simulated darkness makes our VE more arousing. However, we found no effects of a fictitious real-world follow-up task on the emotional response to the VE.

## Effects of personal relevance on mood

The emotional state of the participants was measured twice with the Positive and Negative Affect Scale (PANAS): once before the participants had read their instructions (T1) and once after they finished their exploration of the VE (T2). The results are listed in Table 3.

We quantified a mood change as the difference between the PA (Positive Affect) and NA (Negative Affect) ratings obtained at respectively T1 and T2, and represented them by the variables PADIFF = PA(T2)−PA(T1) and NADIFF = NA(T2)−NA(T1). The assumptions of normality (verified by a Shapiro–Wilk test) and sphericity (verified by Levene's test) were satisfied for the variables PADIFF and NADIFF.

First we investigated whether the experience of a dark VE affects mood. A one-sample *t*-test showed that there was no significant difference between both the PA and the NA ratings obtained at respectively T1 and T2 for participants who explored the nighttime VE without an additional assignment (i.e., both PADIFF and NADIFF did not differ significantly from zero: NADIFF $t(11) = .789$; n.s. PADIFF $t(11) = −.722$; n.s.). Thus, it appears that the exploration of the dark VE did not affect the mood of the participants.

Next we investigated whether ambient darkness in the VE in combination with personal relevance affects mood. In addition, we compared the PADIFF and NADIFF measures between participants who explored the nighttime VE with the fictional follow-up real world nighttime assignment to the ratings given by participants without this follow-up
Table 4 The mean and standard deviation of the ratings on the Igroup Presence Questionnaire (IPQ).

| Simulated lighting | Fictitious task | GPR | | SPR | | INV | | REA | |
|---|---|---|---|---|---|---|---|---|---|
| | | M | SD | M | SD | M | SD | M | SD |
| Daylight | No task | 0.83 | 1.47 | 0.35 | 0.95 | 0.54 | 1.26 | −0.33 | 0.76 |
| | Daylight | 0.58 | 1.56 | 0.48 | 0.99 | 0.58 | 1.01 | −0.29 | 0.77 |
| | Darkness | 1.42 | 1.38 | 0.63 | 1.22 | 0.35 | 1.36 | −0.4 | 1.07 |
| Darkness | No task | 1.17 | 1.19 | 0.78 | 0.97 | 0.58 | 0.96 | 0.02 | 0.70 |
| | Darkness | 0.42 | 1.44 | 0.62 | 1.16 | −0.15 | 1.19 | −0.25 | 0.93 |
| | Daylight | 1.00 | 0.95 | 0.97 | 0.90 | 1.00 | 0.78 | −0.1 | 0.88 |

task. The results indicate that neither the PADIFF ($F(1, 22) < 1.872$, n.s.) nor NADIFF ($F(1, 22) < 1$, n.s.) measures differ significantly between both groups. Thus, our hypothesis (H3) that participants who explored the nighttime VE with the fictional follow-up real world nighttime assignment experience (H3a) a larger decrease in *Positive Affect* and (H3b) a larger increase in *Negative Affect* than participants without this information, is not confirmed by the present results.

Summarizing, we found no effects of a fictitious real-world follow-up task on the emotional state of the participants.

### Effects of personal relevance on presence

Table 4 lists the ratings for each of the factors on the IPQ questionnaire. The reliability of each IPQ factor except GPR (*General Presence*, which consists of only a single item) was tested by calculating the Cronbach's alpha. SPR (*Spatial Presence*) has a good internal consistency ($\alpha = 0.80$ for 5 items). The factor INV (*Involvement*) has a lower but still quite acceptable consistency ($\alpha = 0.72$ for 4 items). The factor REA (*Realism*) has a low reliability ($\alpha = 0.45$ for 4 items).

All factors except REA score mostly moderately positive (i.e., slightly higher than the neutral score). Since there were no outliers and since the assumptions of normality and homogeneity were satisfied we used a two-way ANOVA to further analyze the results from the IPQ. The analysis shows that participants with a follow-up assignment in the real world did not experience a significantly different level of GPR, SPR, INV or REA ($F(1, 70) < 1$, n.s.) than participants without such an assignment. Thus, our hypothesis (H4) that participants with a real world follow-up assignment experience a higher degree of presence in the VE than participants without this information is not confirmed by the present results.

Summarizing, the participants experienced only a minimal degree of presence and involvement in most conditions, while the perceived realism of the simulation was somewhat less than neutral. We found no effects of a fictitious real-world follow-up task on the degree of presence experienced by the participants.

### Fear of darkness in the real world

The results listed in Table 5 show that the participants report that in real life they are typically less at ease at night than in daytime. At night they report to be less proficient at

**Table 5** Results of the navigation and orientation questionnaire.

| Statements | M | SD |
|---|---|---|
| I'm very well able to find my way in an unfamiliar environment. | 0.25 | 1.60 |
| I'm very well able to find my way in a familiar environment at night. | 1.39 | 1.51 |
| I'm very well able to find my way in an unfamiliar environment at night. | −1.00 | 1.51 |
| I can orientate very well in the dark. | −0.15 | 1.32 |
| I can orientate very well in daytime. | 1.31 | 1.35 |
| I dare to walk by myself in an unfamiliar environment in daytime. | 2.38 | 1.03 |
| I dare to walk by myself in an unfamiliar environment at night. | −0.32 | 1.54 |
| I feel uncomfortable in the dark. | −0.19 | 1.55 |

finding their way in an unfamiliar environment than in a familiar environment (2nd and 3rd statement). They claim that their orientation capability is better in daytime than in the dark (4th and 5th statement). When walking alone in an unfamiliar real environment they are more afraid in darkness than in daytime (6th and 7th statement). These findings agree with previous reports that young females are typically more afraid in the dark when they are alone and in an unfamiliar environment (*Warr, 1990*). Hence, the participants in this study are comparable to populations used in earlier studies in the sense that they feel less comfortable in darkness in real life.

A Shapiro–Wilk test showed that the independent parameters (lighting condition and task) were not normally distributed. We therefore used a non-parametric Kruskal-Wallis test to analyze the results. The results showed that there were no significant differences between each of the size experimental groups on any of the eight statements used to measure fear of darkness in the real world. This implies that the participants were appropriately randomized over the experimental conditions with respect to their fear of darkness in the real world.

## Game and navigation experience

More than half of the participants ($N = 44$) did not play 3D computer games, while the rest only played *very occasionally* ($N = 14$) or *sometimes* ($N = 13$). Only one participant played 3D games *frequently*. Virtual environments were not used for other activities than gaming by 66 (83%) participants. The remaining 12 participants used virtual environments for other purposes only *very occasionally* or *sometimes*. Thus, the sample used in this study probably had not much game and navigation proficiency.

## CONCLUSIONS AND DISCUSSION

This study investigated whether increased personal relevance of a desktop VE (induced through a fictitious assignment to visit the corresponding real world environment at night) enhances the negative appraisal of the nighttime VE (H1), intensifies both short term (emotions; H2) and longer lasting (mood; H3) negative affective feelings, and enhances the experienced degree of presence in the VE (H4).

In agreement with previous studies we found that simulated darkness does indeed negatively influence the affective appraisal of a desktop virtual environment: our nighttime

version of the VE was experienced as significantly less cosy and more tense than its daytime counterpart. Simulated darkness also made the VE more arousing. However, we found no indications that personal relevance of the simulation intensifies these effects. Also, in general we found no effect of personal relevance on respectively the affective appraisal of the VE, short term or longer lasting affective feelings of the participants, and the degree of presence that participants experienced in the VE. Thus, the present results did not confirm any of our four hypotheses. Our finding that the nighttime VE was appraised as slightly more *Cosy* with a real world nighttime follow-up assignment compared to no assignment is rather unexpected and contrary to our expectations (hypothesis (H1a)). Maybe the follow-up assignment stimulated the participants to perform a more detailed inspection of the VE, which may have resulted in the impression that the dark environment was probably not so frightening as it appeared during an initial or more superficial inspection.

It seems that darkness has only a small effect on the affective appraisal of an outdoor nighttime scene simulated on a desktop system. This is in contrast to the effects that are typically reported in the literature for similar real world environments. This limits the value of desktop VEs as tools to assess and evaluate the effects of ambient lighting on human feelings of fear. Further research comparing human behavior in—and affective response to—real environments and their virtual counterparts in different lighting conditions is required to establish the reasons for this discrepancy. Until now such studies are scarce (e.g., *Bishop & Rohrmann, 2003*), possibly due to the many practical problems and confounding factors that occur in real world research. For example, some parts of the real world may be difficult or even dangerous to access during darkness. In addition, dynamic environmental elements like moving traffic, clouds, birds and water movement can influence the affective qualities of the scenes, and give them different meanings to different individuals and groups (*Houtkamp, 2012*).

## Limitations of the present study

This study has the following limitations.

The size of the experimental sample was limited. However, from an applied point of view, effects that do not reach significance with group sizes in the order of 10 or more participants are of limited applied relevance, especially when desktop VE's are used to evaluate situations with personal relevance.

One issue concerns the sensitivity of the instruments that are currently available to measure the affective appraisal of environments (e.g., such as the pleasure-arousal scales of *Russell & Pratt, 1980* and the atmosphere metrics of *Vogels, 2008a*, that were used in this study). While these instruments cover all aspects known to determine the emotional response to environments, they do not appear sensitive enough to distinguish responses to subtle effects or differences in the appraisal of environments (especially virtual environments: *Houtkamp, 2012*). Hence, these scales require further refinement to make them suitable to assess the validity of virtual environments for visualization purposes.

The degrees of presence and involvement experienced by the participants in this study were rather low. There may be several reasons for this finding. First, the perceived realism of the simulation was somewhat less than neutral, which may have diminished

the participants' sense of presence. In addition, the virtual environment represented a low level of entrapment and concealment, and therefore may not have been potent enough to induce strong affective feelings, even in darkness. Finally, most participants did not have much game and navigation proficiency. As a result, their navigation through the VE may have required additional cognitive resources and may have distracted their attention from the VE, thus preventing them from achieving a stronger sense of presence (*De Kort et al., 2003*).

The presence of the experimenter may have had a reassuring effect (social presence) on the participants, thus limiting any possible negative emotional effect of the VE. In addition, it may have distracted the participants from their task, thereby reducing their sense of presence.

Maybe the participants where not really convinced that they had to perform a follow-up assignment. The fact that there was no difference observed for the SAM ratings at T1 between participants who did and those who did not receive a follow-up assignment suggests that this information did not cause significant concern. Future studies should make the follow-up assignment more believable to achieve the expected effects, and should include a manipulation check to ask participants how convinced they actually were about the task.

All experiments in this study were performed during daytime. The participants navigated the nighttime virtual environment in a room that was darkened by covering the windows and turning off the light. A recent study investigating the effects of 'night' and 'darkness' on feelings of fear found that the effect of fear stimuli is modulated by the actual time of day (circadian or day-night cycle): fear-provoking stimuli trigger more intense responses in the nighttime condition than in the equivalent daytime condition (*Li et al., 2015*). Thus, it seems that night amplifies fear signals and increases fear responses. This facilitation of nighttime threat responses may reflect an evolutionarily adaptive mechanism for an efficient processing of threat-related stimuli to avoid danger. Although the size of this effect is only small to medium, a replication of the current study in nighttime conditions might amplify the present results. To obtain ecologically valid results future simulation studies should therefore take the day-night cycle into account by performing measurements during a timeframe that corresponds to the simulated time of day (i.e., synchronize actual and simulated time by presenting simulated nighttime conditions at night and simulated daytime conditions during the day).

## ACKNOWLEDGEMENTS

We thank our reviewers for their many suggestions and comments which clearly helped us to improve the quality of our paper.

### Funding

This research has been supported by the GATE project, funded by the Netherlands Organization for Scientific Research (NWO) and the Netherlands ICT Research and

Innovation Authority (ICT Regie). The funders had no role in study design, data collection and analysis, decision to publish, or preparation of the manuscript.

## Grant Disclosures

The following grant information was disclosed by the authors:

Netherlands Organization for Scientific Research (NWO).

Netherlands ICT Research and Innovation Authority (ICT Regie).

## Competing Interests

The authors declare there are no competing interests.

## Author Contributions

- Alexander Toet analyzed the data, wrote the paper, prepared figures and/or tables.
- Joske M. Houtkamp conceived and designed the experiments, analyzed the data, contributed reagents/materials/analysis tools, wrote the paper, reviewed drafts of the paper.
- Paul E. Vreugdenhil conceived and designed the experiments, performed the experiments, analyzed the data, contributed reagents/materials/analysis tools, wrote the paper, reviewed drafts of the paper.

## Human Ethics

The following information was supplied relating to ethical approvals (i.e., approving body and any reference numbers):

The study protocol was approved by the TNO Research Ethics Committee, as confirmed in an approval letter d.d. January 2011.

## Data Availability

The raw data has been supplied as Data S1.

## Supplemental Information

Supplemental information for this article can be found online at http://dx.doi.org/10.7717/peerj.1743#supplemental-information.

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
