# Peer review of "Effects of personal relevance and simulated darkness on the affective appraisal of a virtual environment"

_PeerJ, doi:10.7717/peerj.1743_

## Round 0.1 · original submission · Major Revisions

This submission has received two expert reviews, and both reviewers see merit in the work, although some revisions are required before it can be accepted.

When preparing your resubmission, please be sure to:

- provide a comprehensive introduction for the research, including the specific hypotheses being tested.
- clarify how the research advances the field, in particular within the context of related work by Kim et al 2014 and Boomsa & Steg 2014. Elaborate a bit more on the related work to help better situate the contributions of the current work.
- clarify the experimental methods. Reviewer 1 has many concrete points that should be addressed. In addition, please describe what the ambient light level was in the room while the VR was simulating darkness. This could be a very important point. Were participants dark-adapted? How might their adaptive state have affected their perception?
- clarify the results. Reviewer 1 again has many points that should be addressed. In particular, are the assumptions met for the ANOVA?
- in your concluding section, please clarify how the results of the experiment relate to the hypotheses being tested, and also explain what are the wider implications of the research.

Reviewer 1 ·

Basic reporting

The introduction is well-written but very concise and would benefit from more elaboration – also to clarify where the research sits within the field.

The authors state that it is not yet known how and to what extent simulated darkness in desktop VEs affects the user’s affective appraisal of the represented environment (line 70-71). But recently more research has been done on this topic, some of which is mentioned by the author but not elaborated upon, e.g. Kim et al., 2014; Boomsma & Steg, 2014 JEP.

Experimental design

A clear research question is lacking, this would help the reader in understanding the main knowledge gap identified by the authors. I think the investigation into personal relevance is the important question here, as hypothesis 1 has already been investigated (and replicated) in other studies. Thus, the focus on personal relevance should be clearer throughout the introduction and needs a more detailed literature background.

With regards to the method section:
- The study consisted of different elements and stages of measurement, and I found myself having to go back and forth between the different sections to understand the design of the study. I would advise including a table to summarise the design and indicate when each variable was measured. Also, bringing the participants, experimental design and procedure section forward (before the materials) would help in the readability of the method section.
- The material section includes a lot of measures and it is not always clear how each variable relates to the hypotheses stated in the introduction. The authors could clarify the structure by dividing the variables into control and experimental measures.

More detailed remarks:
Line 168: It is unclear to me why this measure is included in this study. Could the authors elaborate on this?
Line 179 – 202: Why do the authors expect the follow-up assignment to affect short-term emotional state and exposure to the VR environment to affect longer lasting emotional states?
Line 183: which three dimensions?
Line 215: Map drawing is not a measure, but should be part of the materials
Line 253: How were these volunteers recruited?

Validity of the findings

I think the results section is the main area where significant improvements need to be made. As an overall comment: the sample size is small, with only 12 participants in each condition. I therefore doubt whether the assumptions for ANOVA are met. Could the authors comment on this? If the assumptions are violated, please use non-parametric tests instead.

Reports on the results is limited and lacking in details, e.g. when referring to main and interaction effects, the authors need to explain what exactly is being tested, and when using an ANCOVA it needs to be clear what the covariate is and why the results are being controlled for this variable.

The results need to be described in a more meaningful manner, so it is clearer for the reader what the findings and implications are. For instance, in the section on environmental appraisal:

- Instead of line 293 “cosiness scored significantly lower for the nigh-time environment than for its daytime equivalent”, I would suggest providing a more meaningful account of the results (i.e. what does a lower score of cosiness mean?) such as: participants rated the night-time VE as less cosy, intimate and safe compared to the day-time VE.
- line 295: no significant effects on…. were observed
- line 296: no interaction effects between….. were found

Similar comments apply to the other sections in the results.

At the moment, the authors do not refer back to the hypotheses in the results section. Also, the readability of the section could be further improved if the results were ordered in line with the hypotheses, with the important results discussed first followed by the additional, control measures.
Some more specific comments on the results:
- Line 333: is this the T1 or T2 measure of the SAM?
- Line 336: is this for both the day and night-time VE?
- Line 357: I’m assuming PA is Positive Affect – but please state this clearly in the text
- Line 374-375: which finding is this statement based on?
- Line 375-278: unclear sentence, please rephrase
- Line 383: why MANOVA, what are the dependent variables?

For the discussion:
Summary of findings is clear, but some clarification needed:
Line 401: ‘The VE experience itself’ – thus, irrespective of lighting condition?
Line 404: when was this open question asked, what was asked exactly and in which context?
Line 409: weren’t there three fictitious task conditions?
Line 417 – Line 421: this is a crucial finding, please elaborate on this more, what else could explain the finding that positive affect was higher in the night-time VE for this group
Line 423-428: Other studies have found a more consistent effect of VE environments on feelings of safety, explain more clearly why this might not have been replicated here. Refer to the limitations for example, tie these sections in more.
Line 427: please elaborate, which practical problems/confounding factors are you referring to
Line 430: The small sample size should also be discussed as a limitation

A conclusion paragraph at the end is missing – what are the wider implications of your research?

·

Basic reporting

This study evaluates the affective appraisal of a desktop virtual environment (VE) when experienced under daylight conditions or under simulated darkness. Since VEs are often used to evaluate lighting or architectural aspects, e.g. in the context of city design, it is important to understand whether findings from such VEs in fact apply to the real world.

Generally, I found the paper well written and easy to follow. The experimental details were clear, although as a reader not coming from psychology I would have liked to have a bit more detail regarding the various questionnaires used to assess emotional state and affective appraisal. Although I can always refer to the provided references, a bit more detail would have been useful for completeness.

I can see the importance of clearly understanding to what extent VEs can be used to study how changes in real environments might affect people. Although I imagine in immersive VEs the conclusions might transfer more easily, immersive setups are not widely available yet so desktop VEs are still likely to be commonly used for evaluations of real lighting and environments. Additionally, studies such as this can provide useful insights to game designers. There are however several other studies referenced that seem to explore similar hypotheses. I can see that the present study explores a different VE and additionally evaluates the effect of personal relevance, however I feel that given the previous work described, it would be good to have a clearer statement of what the present study offers that previous work has not covered.

Experimental design

In terms of experiment design, execution and analysis I have very little to say. The study seems very thorough and detailed. I have a few minor questions/remarks, mostly in terms of aspects that might explain the relatively low presence:
- In terms of the setup, how far were participants from the screen? How loud was the sound of the VE? Were others in the room with the participant? Also, the ambient light was at what level? I imagine several of these things can play a role in terms of presence in the VE, which might affect the results.
- Although I can understand the navigation aspect (i.e. that if they have trouble navigating they might pay more attention to the navigation aspect and not be immersed in the VE), I'm not sure that the self evaluation there is necessarily sufficient to assess their ability to navigate. Wouldn't an actual navigation task of some form give a more accurate view as to whether participants can navigate or not?
- How was the VE rendered? How realistic was it? How believable was it seen as by the participants? I imagine that the perceived realism of a VE will influence how immersed participants might be and therefore how strong their emotional responses might be to aspects such as darkness.

Validity of the findings

I have no issue in this area. The findings of this study to me seem valid and valuable. The study confirms previous findings and adds new knowledge by exploring the additional aspect of personal relevance. The conclusions of the study link well with the initial hypotheses as well as with findings of previous work.

Additional comments

Overall I think the paper clearly defines a problem and studies it in sufficient detail to add some knowledge in the field. I can see that it can lead to several follow up studies, particularly focusing on the effects of realism of the VE and surrounding aspects in the viewing environment (e.g. lighting in the real room, people present, ambient noise, etc), which might help optimize viewing conditions for desktop VE tasks such that they correspond more closely to real viewing conditions.

The authors could add a bit more detail regarding their chosen questionnaires for non-expert readers but I would say this is an optional addition, the paper is sufficiently complete and provides clear references anyway.

---

## Round 0.2 · Minor Revisions

Both reviewers are happy to see that the manuscript has been substantially improved, and are looking forward to seeing the results published. Prior to publication, however, there are a few additional minor revisions that need to be made. Once completed, these minor revisions can be verified by the AE without the need for an additional round of review.

Reviewer 1 ·

Basic reporting

The authors have significantly rewritten the paper, which I feel has improved it. Especially the introduction section sets out the proposed research a lot clearer, and the research questions and hypotheses are more focused.

However, I do feel a few minor changes and clarifications are needed before the paper is ready for publication.

Comments on the Introduction
line 75-77 and line 79-81 are very similar, please rephrase.
line 95 "differences in affective appraisal were small" please clarify which differences are referred to here.
line 138: at this point this is still an hypothesis (this could explain why these studies found these effects) so be careful not to present it as a finding

Also - could the authors check the reference list - there are some inconsistencies in the reference style etc.

Experimental design

Comments on the Method
line 184 'in two conditions' - this needs clarification. The study had 6 conditions, in 4 of these participants received instructions about a follow-up tour.
line 208-214: I wonder if it is necessary to include the websites here, I don't feel this adds anything
line 246-247 this definition is unclear, please rephrase
line 259: make sure the use of terms is consistent, so I wonder if 'affective appraisal' should be the heading here?

Also, I think it would be helpful if a definition of affective appraisal is included at the start of the manuscript (e.g. line 37-39) rather than only in the Method section - as this is a key variable.

line 289 right bracket seems to be missing?
line 323 presence is included in a hypothesis so should be part of the 'experimental measures' section and not the control section

To improve the readability of the method section I would recommend changing the order of the method to:
- participants
- experimental design
- procedure
- materials/measures
- data collection & analysis

Validity of the findings

Comments on the Results
line 427 which analysis does this test result refer to?
line 433 please report test results here
line 437 acknowledge here that the expected effects were not found - but an effect was found for cosiness
line 447-462 could the authors clarify in this section whether these analyses concern the T1 or T2 measurement, or a difference score?
line 523-533 this section should also include statistical comparisons to underpin the results

Comments on the Discussion
line 557 this conclusion does not seem to follow directly from the results? the study presented in the paper did not compare a VE to a real environment. I think the conclusion needs to focus more directly on what the implications of the current study are.
line 574 the problem with a small sample size is that it can limit the generalisability of the results. This is further complicated by the fact that not a lot is known about the sample used for this study (e.g. educational level/profession etc). Please comment on this in the discussion. I don't think the current line of reasoning is very strong.

Also, I feel the finding that the nighttime VE with nighttime assignment is seen as more cosy that without the assignment should be addressed in the Discussion.
- what could be the reason for this finding?

Furthermore, I wonder whether one reason why the assignment did not lead to any of the expected effects was because participants did not believe the tour would actually happen? The study did not include a manipulation check to ask participants about this, so the authors won't be able to say whether this was the case or not. But I think this needs to be addressed in the Discussion, maybe future studies should make the follow-up assignment more believable to achieve the expected effects?

Additional comments

I can see a lot of work has gone in the revision and I think the paper has improved a lot. The question around personal relevance is intriguing, I don't think the non-significant findings necessarily indicate that it is not an important factor. It opens up interesting questions for future research!

·

Basic reporting

No Comments

Experimental design

The authors have followed the suggestions of reviewers and have provided additional details and clarifications regarding their experiments. This helps the readability of the paper.

Validity of the findings

No Comments

Additional comments

The authors have considered the reviewers comments and have provided additional details and clarifications. They have refined their analysis and added background that might be necessary for non-expert users.

---

## Round 0.3 · accepted · Accept

Thank you for your careful attention to all of the points raised by the reviewers, especially reviewer 1. I agree with this reviewer that the inclusion of explicit URLs for the Deltares, Dell and Altec Lansing websites is not necessary because, if interested, readers can easily perfiorm Google search on these names - I just tried it and in each case, the desired website came up as the first hit. I do think that in other cases, such as later on for the IPQ, the inclusion of the URL is helpful.